# On the use of in-silico simulations to support experimental design: A case study in microbial inactivation of foods

**Alberto Garre[1][¤], Jose Lucas Peñalver-Soto[1], Arturo Esnoz[1], Asunción Iguaz[1], Pablo S. Fernandez[1], Jose A. Egea [2]***

**1** Departamento de Ingeniería de Alimentos y del Equipamiento Agrícola, Instituto de Biotecnología Vegetal, Universidad Politécnica de Cartagena (ETSIA), Paseo Alfonso XIII, Cartagena, Spain, **2** Centro de Edafología y Biología Aplicada del Segura, Campus Universitario de Espinardo, Murcia, Spain

¤ Current address: Food Microbiology, Wageningen University & Research, Wageningen, the Netherlands
* jaegea@cebas.csic.es

## Abstract

The mathematical models used in predictive microbiology contain parameters that must be estimated based on experimental data. Due to experimental uncertainty and variability, they cannot be known exactly and must be reported with a measure of uncertainty (usually a standard deviation). In order to increase precision (i.e. reduce the standard deviation), it is usual to add extra sampling points. However, recent studies have shown that precision can also be increased without adding extra sampling points by using Optimal Experiment Design, which applies optimization and information theory to identify the most informative experiment under a set of constraints. Nevertheless, to date, there has been scarce contributions to know *a priori* whether an experimental design is likely to provide the desired precision in the parameter estimates. In this article, two complementary methodologies to predict the parameter precision for a given experimental design are proposed. Both approaches are based on *in silico* simulations, so they can be performed before any experimental work. The first one applies Monte Carlo simulations to estimate the standard deviation of the model parameters, whereas the second one applies the properties of the Fisher Information Matrix to estimate the volume of the confidence ellipsoids. The application of these methods to a case study of dynamic microbial inactivation, showing how they can be used to compare experimental designs and assess their precision, is illustrated. The results show that, as expected, the optimal experimental design is more accurate than the uniform design with the same number of data points. Furthermore, it is demonstrated that, for some heating profiles, the uniform design does not ensure that a higher number of sampling points increases precision. Therefore, optimal experimental designs are highly recommended in predictive microbiology.

**Data Availability Statement:** All relevant data are within the paper and its Supporting Information files.

**Funding:** The financial support of this research work was provided by the Ministry of Science, Innovation and Universities of the Spanish Government and European Regional Development Fund (ERDF) through project AGL2017-86840-C2-1-R, as well as the Seneca Foundation through project (20900/PD/18). AG is grateful to the MINECO for awarding him a pre-doctoral grant (Ref: BES-2014-070946). The funders had no role in study design, data collection and analysis, decision to publish, or preparation of the manuscript.

**Competing interests:** The authors have declared that no competing interests exist.

## 1 Introduction

Predictive microbiology is nowadays a basic tool in food safety research [1]. It provides mathematical models whose applications include the prediction of the microbial response to environmental conditions, such as those encountered during storage or food processing [2–4]. Another use of predictive models is inference, where the response to different bacteria is compared in order to, for instance, identify the most resistant bacterial strain to some treatment [5,6].

Most of the mathematical models used in predictive microbiology are parametric models, with unknown parameter values that have to be estimated using experimental data. This requires the definition of an experimental design and carrying out the experiments. Then, a model fitting algorithm is used to calculate estimates for the model parameters. The precision and accuracy of the parameter estimates is critical for the precision of the model predictions [7,8]. In this context, accuracy is understood as model predictions (or parameter estimates) being unbiased with respect to the actual values, whereas precision refers to their spread [9]. Accuracy can be tested by comparing model predictions against experimental observations not used for model fitting (e.g. isothermal versus non-isothermal experiments [10–13]) or using resampling techniques [14]. Precision can be quantified using some measure of uncertainty, for instance, the standard deviation of model parameters.

Because uncertainty and variability are inherent to any microbiological experiment [15], a measure of precision must be reported in every predictive microbiology study and considered for model predictions. Therefore, a probabilistic approach must be followed, where model predictions are expressed as confidence regions, calculated based on the standard deviation of the model parameters [16]. Situations with low uncertainty/variability will be reflected in small standard deviation of model parameters and, thus, narrow confidence regions for model predictions [17].

It is usually anticipated that an increase of the number of data points used for model fitting should improve the precision with which the model parameters are known, reducing their associated standard deviation. Several studies have shown that this can also be achieved through the use of alternative experimental designs, reducing the uncertainty of the model parameters with the same experimental work [18–20]. This has led to the development of the field denominated Optimal Experiment Design (OED), which tries to find the most informative experimental setting under some constraints such as number of data points [21]. It has been successfully applied to several applications in food science, such as the characterization of microbial growth and inactivation [8,17,22–27].

Different approaches have been applied to date for OED. One of the most extended is based on the properties of the Fisher Information Matrix (FIM), which measures the amount of information that a vector of observable random variables carries about a vector of unknowns (response variables) [28]. Therefore, the FIM can be used to assess the efficacy for parameter estimation of different experimental designs and for obtaining optimal ones. It has several relevant properties for parameter estimation. For instance, according to the Cramer-Rao theorem, the inverse of the FIM is a lower bound of the covariance matrix of the model parameters [29]. Hence, the inverse of the determinant of the FIM can be used as an estimate of the volume of the confidence hyper-ellipsoid [30]. This property leads to the so-called D-optimality criterion for OED. The superiority of D-optimal designs with respect to uniform designs has already been demonstrated in several works (e.g. [17,31,32]).

The model parameters of models used in predictive microbiology usually have a biological meaning. For instance, the D-value describes the treatment time required to reduce the microbial count a 90% [33]. Model parameters estimated under certain conditions are commonly

used to, for example, infer the effectiveness of a treatment [6,34]. Therefore, in many situations, the objective of experiments designed in the context of predictive microbiology is not prediction but the estimation of model parameter with enough precision (standard deviation) that enables accurate inference. Despite the advances in OED, there are still some open questions when it comes to designing such experiments. For instance, the number of sampling points is commonly decided based on previous experience. As a result, there is a high risk that the number of sampling points is excessive, leading to unnecessary experimental work, or too low, which would require posterior repetitions of the experiment. In this work, we explore the application of numerical techniques to reduce this uncertainty. We propose two complementary methodologies, the first one based on the properties of the FIM and the second one based on Monte Carlo simulations. Although both methods are usually applied to compare between different designs, here we illustrate how they can be used to aid in the decision process during the first stages of the experimental design. We describe their mathematical basis and illustrate how they can provide valuable information that may reduce the uncertainty of the experimental design (e.g. in the selection of the number of sampling points). For this, we analyze a case study related to dynamic microbial inactivation. Nevertheless, the applicability of these methods is not restricted to this case and could, in principle, be applied to any problem in the context of predictive microbiology.

## 2. Materials and methods

### 2.1 Simulated experimental setting

In order to better illustrate the methodology proposed in a controlled setting, the work has been done *in silico* to avoid uncertainties introduced by the experimental methodology. *Listeria monocytogenes* has been selected as model microorganism. According to the results of Garre et al. [17], it has been assumed that the Bigelow model [33,35] is able to describe the non-isothermal survivor curve of this microorganism. This model is defined for non-isothermal conditions in Eq (1), where $D_{ref}$ stands for the D-value (time required for ten-fold reduction of the microbial load) at the reference temperature, $T_{ref}$. This temperature has no biological meaning, but can influence parameter identifiability [36,37]. The sensitivity of the D-value to temperature changes is quantified by the z-value, $z$, which states the temperature increase required for ten-fold reduction of the D-value.

$$\frac{d \log N}{dt} = -\frac{1}{D_{ref} 10^{-\frac{T - T_{ref}}{z}}}$$ (1)

Three different non-isothermal processing treatments have been tested. They have been selected because they are typical profiles used to characterize microbial inactivation. All of them begin between 30°C and 45°C, and reach a maximum temperature between 60 and 65°C. However, the heating rate and the shape of the profile differ. Profiles named A and C are monotonically heating profiles with a heating rate of, respectively, 0.5°C/min and 10°C/min. Profile B is a biphasic profile, with a heating phase at 1°C/min between 30 and 60°C, followed by a cooling phase with the same cooling rate until the initial temperature is reached. Fig 1 illustrates the three temperature profiles. Although there is experimental evidence indicating otherwise [12], in order to simplify the simulations, it has been considered that the model parameters estimated by Garre et al. [17] for *L. monocytogenes* under isothermal conditions are suitable to describe the microbial inactivation for the three profiles tested ($D_{57.5°C}$ = 3.9 *min*, $z$ = 4.2°C). Note that, despite its biological significance, this hypothesis is irrelevant to the purpose of this article, as the model parameters are used to evaluate the precision of the experimental designs. The expected bacterial response is shown in Fig 1 as a dotted line.

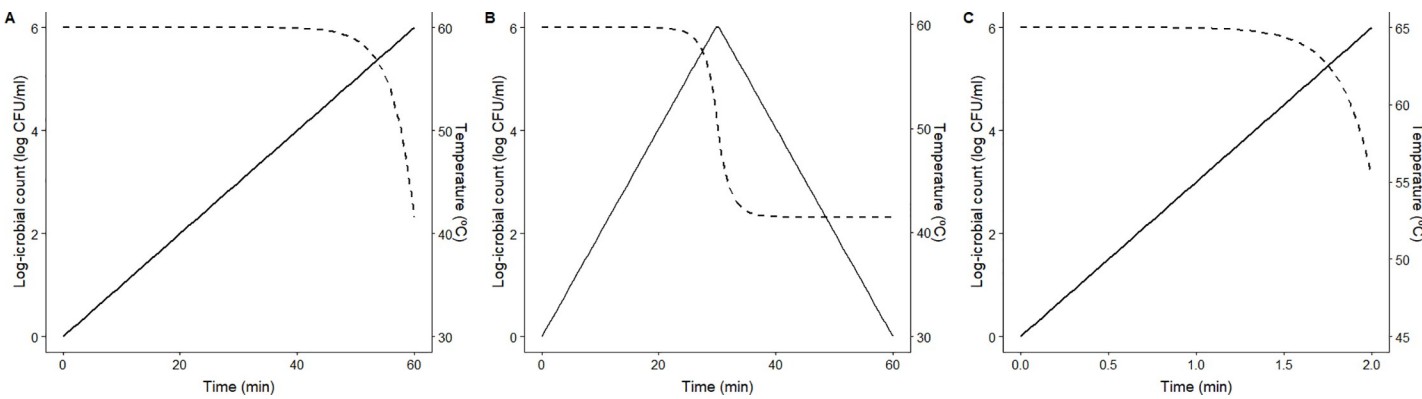

**Fig 1.** Thermal profiles (A, B and C) analysed as a case study (-). Expected survivor curve for each profile (—).

## 2.2 (Optimal) experiment designs considered

The type of data sampling of the experimental designs for dynamic microbial inactivation has been compared in this study. The first one is the "classical" uniform design, where sampling points are uniformly distributed through the duration of the experiment. The second is the approach to OED for dynamic microbial inactivation proposed by Garre et al. [17] and implemented in the *bioOED* package for R(https://CRAN.R-project.org/package=bioOED). This methodology is based on the optimization of a measure of the Fisher Information Matrix (FIM), an approach commonly followed for OED [14,31,38], adding a penalty function to avoid infeasible experimental designs (with too close sampling points).

The functions included in the *bioOED* package allow to apply different optimality criterions of the FIM. In this work, the D-optimal criterion, which consists in the maximization of the determinant of the FIM, has been selected [39]. Other criteria have been suggested to identify OEDs. A popular alternative is the E-criterion, which tries to minimize the maximum uncertainty in parameter estimates (in the case studied here, the uncertainty of the parameter with the highest error) [31]. Because this article tries to illustrate how computational methods can be used to aid experimental design and because this criterion has already been applied in a similar problem [17], this study is limited to the results of the D-criterion. A comparison between the precision of different designs is left for future work.

Accordingly, the OED is reduced to the optimization of the function shown in Eq (2), where $\frac{\partial y}{\partial p}(t_i)$ represents the first derivative of the outcome variable (the log-microbial count) with respect to each model parameter ($D_{ref}$ and $z$ in the Bigelow model) evaluated at each sampling point, $t_i$. $Q$ is a weight matrix, which has been set to the identity matrix. A negative penalization term, $P(t_i)$, (explained below) is added to the expression to be maximized to avoid impossible solutions from the practical point of view.

$$\max_{t_i}\left[\det\left(\sum_{i=1}^{N}\left(\frac{\partial y}{\partial p}(t_i)\right)^T \cdot Q \cdot \left(\frac{\partial y}{\partial p}(t_i)\right)\right) - P(t_i)\right] \tag{2}$$

The methodology proposed by Garre et al. [17] to avoid infeasible designs, where sampling points are too close, has been applied. This approach introduces a penalty function ($P(t_i)$) in the optimization problem (Eq 2) that penalizes designs with sampling points closer than a threshold ($t_{min}$). $P(t_i)$ is a barrier penalty function with the algebraic form shown in Eq (3). In

this study, the minimum time between samples, $t_{min}$, has been set to three seconds.

$$P(t_i) = \begin{cases} \left(2 - \dfrac{t}{t_{min}}\right) \cdot 10^5, & t < t_{min} \\[2ex] \dfrac{1}{e^{t - t_{min}} - 1}, & t \geq t_{min} \end{cases} \tag{3}$$

For both the optimal and uniform schemes, designs have been generated for a different number of sampling points (starting from ten), to evaluate how increasing the sample size affects each experiment design.

## 2.3 Numerical simulation of in-silico experiments

In the absence of experimental error, under the hypothesis that the Bigelow model (with the selected parameter values) is able to describe the microbial inactivation, the microbial counts shall not deviate from the model predictions. However, the different sources of uncertainty and variability would cause a scatter of the experimental observations. In this work, it has been considered that these sources of error are additive and uncorrelated. Hence, their effect on the measurements has been modelled as a white noise ($\varepsilon$) with variance $\sigma^2$ as shown in Eq (4), where $\hat{N}_i$ stands for the microbial count predicted by the Bigelow model and $\tilde{N}_i$ is the simulated observation.

$$\log \tilde{N}_i = \log \hat{N}_i + \varepsilon_i; \ \varepsilon \sim N(0, \sigma^2) \tag{4}$$

Accordingly, the microbial counts that would be observed in the laboratory have been simulated for the different experimental designs. The microbial count predicted by the Bigelow model at the time points used in the experimental design has been calculated using the functions implemented in the *bioinactivation* R package [40,41]. At each time point, three different normally distributed random numbers have been generated, to simulate three repetitions of the experiment.

## 2.4 Model fitting and data analysis

The simulated experimental results have been fitted using the non-linear regression functions implemented in the *bioinactivation* package [40,41]. Several experiments have been simulated for each design, and the distribution of the parameter estimates for each one of them has been analysed. The number of simulations has been increased until the mean and standard deviations of the distribution of model parameters has converged to stable values. The functions required for the computational work have been implemented in R version 3.4.3 [42] and are provided as supplementary material (S1 Code and S2 Code).

## 3. Results and discussion

### 3.1 Description of the optimal experiment designs identified

Fig 2 give a qualitative description of the designs identified as optimal for each one of the profiles analyzed (Fig 1), as well as on how they vary when the number of samples is increased. The x-axis represent the sample space (the duration of the experiment) and the height of the bars the number of samples that are located at a given location. The total number of sample points is represented by the colour of the bar (see legend in Fig 2). The bars corresponding to different number of points are stacked on top of each other, so the total height of the bar is a representation of the use of a time point across a different number of samples.

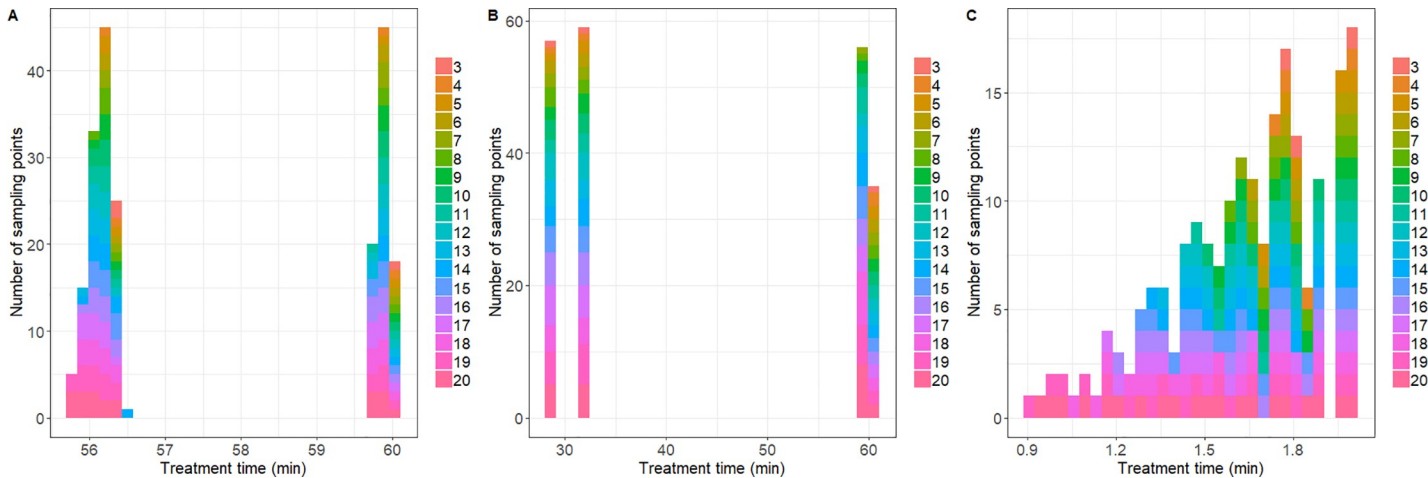

**Fig 2.** Frequency bar plot illustrating the OED calculated for the three profiles analyzed (A, B and C). The colour of the bar indicates the total number of sampling points.

For every profile analysed, some areas of the sampling space are the most informative and the algorithm tends to locate samples in that location. As expected, the most informative design pattern depends on the shape of the thermal profile. These differences between profiles can be justified based on the local sensitivity functions of each profile, shown in Fig 3. For profile A, samples are located close to the end of the treatment, at approximately 56 and 60 minutes (Fig 2A). As illustrated in Fig 3A, $t = 56$ corresponds to a minimum of the sensitivity function with respect to the z-value and $t = 60$ to a supremum of both sensitivity functions. The algorithm is able to identify these areas and locates the sampling points in a configuration that satisfies the constraint related to the minimum distance between samples. For profile A, due to the large duration of the experiment (60 min) with respect to the minimum time between samples (3s), the restriction can be easily fulfilled. Hence, the frequency plot (Fig 2A) shows a large area without any samples between the two informative areas.

Although profile C has a shape resembling the one of profile A, there are several differences that affect the optimal design pattern. Whereas the optimum design pattern for profile A

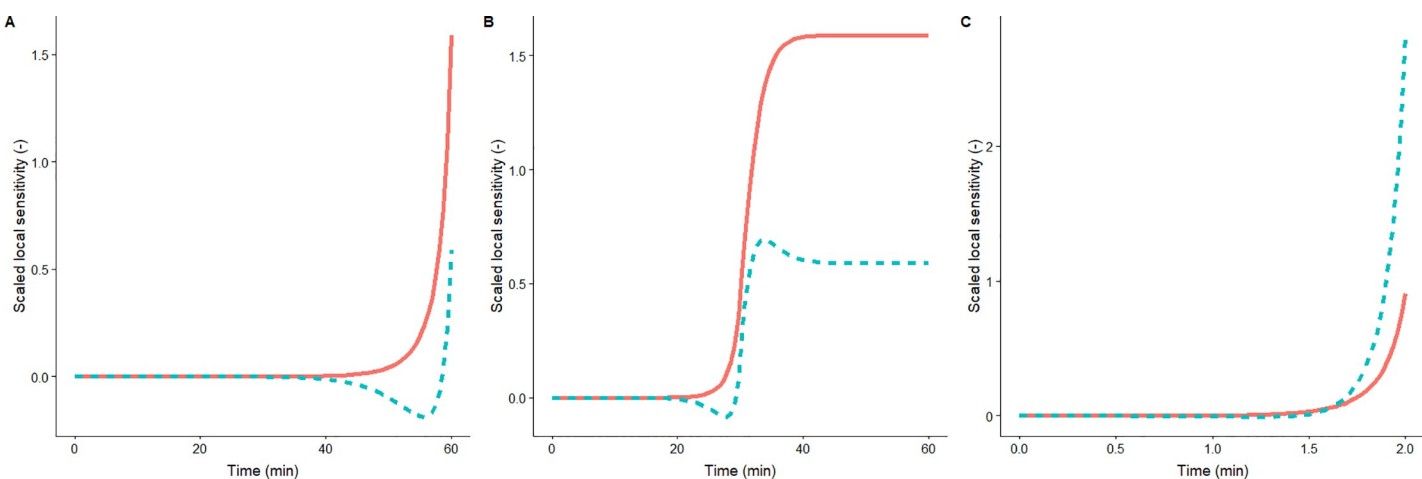

**Fig 3.** Scaled local sensitivity functions of profile A (A), B (B) and C (C) with respect to the D-value (-) and the z-value (—).

distributes the samples in a balanced manner between $t = 56$ and $t = 60$, for profile C the end of the experiment is favoured. This is due to the fact that the sensitivity functions (especially the one of the z-value) grow quickly with temperature for temperatures about the reference one. The maximum temperature in profile C is 65˚C, 5˚C higher than the one reached in profile A. Consequently, the minimum of the sensitivity function with respect to the z-value is less relevant in profile C than in profile A. Furthermore, the duration of profile C is much lower than the one of profile A (2 min vs 60 min). Therefore, the constraint related to the minimum distance between sampling points is much harder to fulfil and the design is more spread-out.

Profile B has a shape very different to the one of profiles A and C. Hence, the optimal design pattern for this profile (Fig 2B) is very different to the one of the other two profiles. Several samples are located close to the middle of the treatment ($t = 28$ and $t = 32$). Both points correspond, respectively, to a minimum and a maximum of the sensitivity function corresponding to the z-value (Fig 3B). Besides this area, samples are located at the end of the experiment, where the sensitivity with respect of the D-value reaches its highest value.

## 3.2 Methodology I: Monte Carlo simulations

Most mathematical models used in food science have model parameters that have to be estimated using experimental data [1]. In this section, a methodology based on *in silico* simulations is proposed to predict the precision in model parameters (understood as the magnitude of their standard deviations) of an experimental design. It can be used to support experimental designs of processes described by parametric models. Although the Bigelow inactivation model is used in this example, this approach is not restricted to it and is extensible to other inactivation models or even other type of models (e.g. growth models).

As an example, let us design an experiment for the characterization of the inactivation kinetics of *L. monocytogenes* using non-isothermal experiments. The factors to consider for the experimental design are three: (1) the temperature profile to use (among the three selected), (2) the number of sampling points and (3) the location of those sampling points. Although previous studies have proposed algorithms to select optimal profiles [8,27,43], this example will be limited to the study of the three inactivation profiles shown in Fig 1. Two criterions will be set to accept the experimental design: 1) that the estimated value of both model parameters ($D_{ref}$ and $z$) should be unbiased (the expected parameter estimates are equal to the values used for the simulations) and 2) their relative standard deviation lower than 0.1, where the relative standard deviation for parameter $\theta$ is defined as $\sigma_{rel,\theta} = \hat{\sigma}_\theta / \hat{\theta}$, where $\hat{\theta}$ is the estimated value for $\theta$ and $\hat{\sigma}_\theta$ its standard deviation.

Without mathematical modelling analyses and tools it is not possible to assess if an experimental design will be successful at characterizing the microbial inactivation with the desired level of precision. This would require carrying out the experiment, fitting the model parameters and calculating their confidence intervals. If the precision was lower than expected, the experiment might have to be repeated, increasing the number of data points. The use of numerical simulations, which can be used to simulate the probability distributions of experimental results [23] is suggested. Therefore, they can be used to simulate the quality of the model fit that would be obtained for an experimental situation. According to the materials and methods section, the following simulation scheme has been performed in this work:

1. Select a type of experimental design.

2. Simulate the "ideal" microbial concentration at the sampling points according to the Bigelow model.

3. Simulate three repetitions of the experiment adding an experimental error as additive white noise.

4. Fit the model to the simulated inactivation data obtained using non-linear regression.

Based on previous experience in similar conditions [3,6], the standard deviation of the error term has been set to 0.25 log CFU/ml for the numerical simulations. This implies that 95% of the observed log-microbial counts are expected to deviate from the ideal value less than ±0.5 log CFU/ml.

As described, the precision of the parameter estimates have been analysed using Monte Carlo simulations. In order to ensure the convergence of the algorithm, calculations have been repeated for different number of Monte Carlo simulations. Increasing their number beyond 100 simulations had no impact on the results (not shown), so the results obtained with 100 Monte Carlo simulations are reported. In every case, the distribution of the estimated model parameters was centred on the actual model parameters used for the simulations ($D_{57.5} = 3.9$ *min*, $z = 4.2°C$), which indicates absence of bias as required. However, significant differences are observed in the standard deviations estimated using the OED and uniform designs, as well as between the different profiles. For profile A, when a uniform design with ten sampling points was used, the D-value was estimated with the required precision ($\hat{\sigma}_{rel} \leq 0.1$) in 78% of the simulations, and only 7% for the z-value. On the other hand, the OED with ten sampling points provided much better results, attaining the desired precision in every simulation for both model parameters. For profile B, only 42% for the D-value and 0% for the z-value when a uniform experiment design with 10 points was used, and 94% for the D-value and 42% for the z-value when the OED was selected. Regarding profile C, the algorithm failed to calculate the standard deviation (the algorithm did not converge) of the model parameter in most simulations when the uniform experimental design with 10 sampling points was used. This is likely due to identifiability issues of the design. When the OED was used, the z-value was estimated with the desired precision in 99% of the simulations, but the D-value in none of them, possibly because most of the inactivation for profile C occurs in a short time (half a minute) at the end of the experiment. This result demonstrates that some inactivation profiles are more informative than others for estimating the D and z-values, even when an OED is used. Therefore, the shape of the temperature profile should be taken into account when designing experiments for characterization of microbial inactivation under dynamic conditions.

The *in silico* simulations enable the identification of profile A as the most informative among the ones tested. Furthermore, among all the options tested, only the experimental design with ten sampling points (with three repetitions) taken according to an OED for profile A are likely to attain the desired level of precision. Therefore, among the temperature profiles tested in this work, profile A should be the one used in laboratory conditions to characterize the inactivation kinetics of the case studied.

## 3.3 Methodology II: Properties of the FIM

The previous section has illustrated how numerical simulations can be applied to assess the likeliness that an experimental design provides the desired level of precision. However, is the number of sampling points selected optimal or could it be reduced with little impact in the precision of the parameter estimates? This question could be tackled following a similar procedure as the one used above, but Monte Carlo simulations are computationally intensive and the computational requirement to perform all the simulations is rather high. For this reason, a methodology based on the properties of the FIM is suggested in this section, to complement the Monte Carlo simulations. The calculation of the FIM (and its determinant) is less

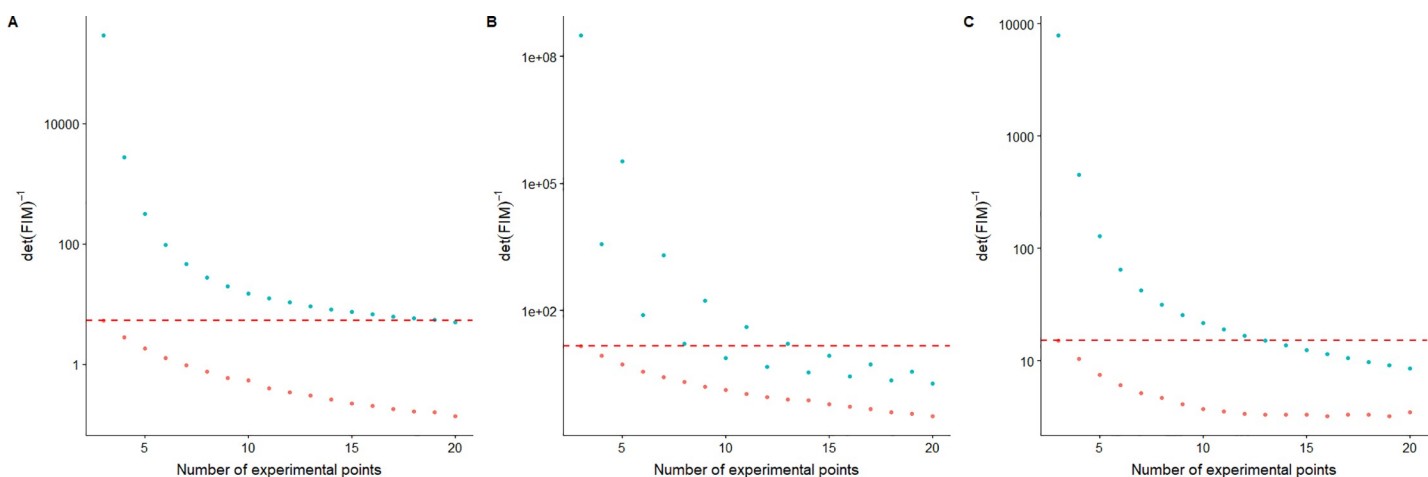

**Fig 4.** Inverse of the determinant of the FIM calculated for each thermal profile (A, B and C) as a function of the number of sampling points for an OED (**red dot**) and for a uniform design (**blue dot**).

demanding from a computational point of view than the Monte Carlo simulations required for the method proposed in the previous section. Hence, it is feasible to calculate it for a large number of experimental designs. On the other hand, the FIM has several shortcomings as an estimator of the variance-covariance matrix (*C*) of the model parameters. According to the Cramer-Rao inequality, the FIM is a bound of *C* under several hypotheses that disregard possible non-linearity in the model [44]. Moreover, the calculation of the FIM and the local sensitivities can be complicated when errors are non-normally distributed, for instance when residuals are heteroscedastic or when they do not follow a normal distribution (e.g. a Poisson distribution). Also, local sensitivity functions can be hard to calculate when the parameter of interest is the variance (e.g. in studies such as [45]). Monte Carlo simulations are more flexible than the procedure based on the FIM are can be applied in such cases with little complexity.

The determinant of the FIM for a uniform and optimal experimental designs has been calculated for the three temperature profiles considered for a varying number of sampling points (three to twenty). The calculated values are illustrated in Fig 4, where the label of each subfigure correspond with the profile names (A and C monophasic profiles with heating rates of 0.5 and 10°C/min, and B biphasic profile). In every case, the determinant of the FIM is lower for the OED than for the uniform design with the same number of sampling points. Therefore, it is to be expected that the OED will result in narrower confidence intervals than uniform designs for the same number of data points. This result is in line with previous conclusions from other authors who also compared uniform and optimal experimental designs for microbial inactivation [17,22], as well as with the results from the previous section. Indeed, (as highlighted by the dashed horizontal lines in Fig 4) 18, 13 and 8 uniformly distributed sampling points are needed, respectively, for profiles A, B and C to obtain the same determinant of the FIM that are obtained for an OED with only three data points (the minimum number of data points required to estimate a model with 2 model parameters with a minimum statistical rigour).

Intuitively, it would be expected that an increase in the number of data points would result in a reduction of the uncertainty associated to the model parameters. However, this is not the case for every situation considered in Fig 4. In the OED, increasing the number of sampling points when their number is low (e.g. from three to four) has a strong positive effect in the determinant of the FIM. This effect diminishes when the number of sampling points is high.

For instance, a plateau is observed for profile C when the number of sampling points is increased beyond ten. Consequently, increasing the number of sampling points beyond these numbers brings little benefit when an OED is used.

Regarding the uniform experimental design, the behaviour observed for profiles A and C is similar to the one observed for the OED. When the number of sampling points is low, adding one more has a strong positive influence in the information provided by the experiment and, consequently, in the precision of the parameter estimates. This impact is progressively diminished as the number of sampling points is increased. Indeed, it plateaus for profile C for more than 12 sampling points. As already discussed above, this is caused by the constraint regarding the minimum distance between sampling points. Due to the shorter duration of this profile, the restriction is hard to fulfil and samples are located in areas that provide little information (Figs 2C and 3C). Therefore, there is a limit in the amount of information that can be extracted for a thermal profile, when a restriction is included to limit the minimum distance between sampling points. For a number of samples beyond this limit, the inverse of the determinant of the FIM may slightly increase due to the numerical error involved in the calculations. Nonetheless, from the results in Fig 4C, it is unreasonable to increase the number of samples beyond 12 for profile C.

Nevertheless, OEDs remain much more informative than uniform designs for the range of data points tested. On the other hand, for temperature profile B, the relationship between the inverse of the determinant of the FIM and the number of sampling points is not monotonically decreasing. Uniform experimental designs with an even number of sampling points are consistently more informative than designs with an odd number of points. The reason for this is the shape of the local sensitivity functions for the Bigelow model in this type of temperature profile. As already illustrated in Fig 3, local sensitivity functions are strongly dependent on the type of temperature profile. Consequently, the location of the most informative sampling points depends on the shape of the profile. OED takes into account the amount of information when the number of sampling points is increased by one. On the other hand, uniform designs simply place the new point according to a uniform partition of the sampling space. Therefore, sampling points may be "moved" from very informative positions to ones where local sensitivities are very close to zero, reducing the information extracted from the system. This is the case for profile C, as illustrated in Fig 4. Hence, not only is the uniform experimental design less informative for dynamic microbial inactivation than OED when the same number of data points is used, but also increasing the number of sampling points in a uniform sampling scheme may reduce the information gained from the experiment.

The differences between the thermal profiles used are not limited to the effect observed in Fig 4C. The value of the inverse of the determinant of the FIM depends on the thermal profile, with profile A taking lower values than profile C, and profile B taking the highest one (note that the subplots in Fig 4 have a different scale to ease comparison between the uniform and optimal designs). Due to the properties of the FIM as estimator of the covariance matrix, the confidence regions estimated from the experiments are expected to be the smallest for thermal profile A. This agrees with the result obtained in the example illustrated in section 3.1, where the standard deviation of the model parameters estimated for profile A was the lowest in comparison with the rest, when the same number of sampling points were used. Therefore, the precision of the parameter estimates is not only affected by the position of the sampling points. Other aspects of the experimental design, such as the thermal profile used for the experiments also affects the parameter estimates. This result is in agreement with the one reported by Van Derlinden et al. [27], who used global optimization to find the most efficient thermal profile for parameter estimation in microbial inactivation.

In order to validate these hypotheses, based on the assumption that the FIM is a good estimator of the covariance matrix, Monte Carlo simulations have been performed for various experimental designs following the methodology illustrated in section 3.1. The results of these simulations are illustrated in Fig 5, where the mean standard deviation of the model parameters (D-value at the reference temperature and z-value) of 100 Monte Carlo simulations is illustrated for the three temperature profiles considered in this study for experimental designs (uniform and optimal) with different number of sampling points. These results confirm the conclusions drawn based on the properties of the FIM (Fig 4). For experimental designs with the same number of sampling points, profile A calculates parameter values with smaller standard deviations for the D and the z values than profiles B and C. Furthermore, the simulations confirm that increasing the number of sampling points does not ensure a greater precision for a uniform experimental design. An increase in the number of data points from eight to nine for profile B has a negative impact in the precision of the parameter estimates. The standard deviation of the D-value is increased by a factor of 36 (121 vs 3.39) and for the z-value by a factor of 13 (18 vs 1.39).

Moreover, the superiority of optimal designs with respect to uniform designs is clearly evidenced for every profile. For profile A, an OED with three measurement points provides a similar precision than a uniform design with 20 measurements for the z-value but a higher precision for the D-value. For profile B, an OED with three sampling points provides more precision than a uniform design with ten sampling points in both parameters. Finally, for profile C, an OED with three sampling points provides more precision than a uniform design with 14 sampling points, also considering both parameters. This result differs slightly from those shown in Fig 4, in which e.g., for Fig 4B an OED with 3 sampling points provides approximately the same information as a uniform design with 8 sampling points. These deviations can be due to the fact that a confidence interval (or the standard deviation) is unidimensional and does not consider the correlation between the parameter estimates, whereas the determinant of the FIM (which is shown in Fig 4) does consider it. Another hypothesis is the effect of model non-linearities that are not considered in the FIM [44]. The position of the sampling points has an impact on the correlation of the parameter estimates [7], so it is to be expected that the correlation between the D-value and the z-value calculated in the uniform design is different from the one obtained using the OED, thus affecting the comparison. Nevertheless, this deviation has a small impact and does not affect the conclusions drawn from the properties of the FIM: (1) the higher precision of the OED, (2) the higher information gained from profile A and (3) that increasing the number of points in a uniform design does not ensure a higher precision.

Both approaches presented here are complementary. The determinant of the FIM can be easily calculated for a large number of experimental designs (including different temperature profiles) and can be used to identify experimental settings that seem more efficient. Then, the Monte Carlo simulations can be used to calculate the accuracy in the parameter estimates expected from the application of this approach. Also, some typical model assumptions (e.g. homogeneity of the residuals) can be checked by means of the Monte Carlo simulations. Both techniques do not require any experimental data, so they can be done before experimental work is carried out. Hence, they can guide experimental design, identifying what inactivation profiles and designs are the more informative, as well as predicting whether an experimental design is likely to provide the desired precision in parameter estimates. Although the case studies have been limited to the Bigelow model and microbial inactivation, the hypotheses used to build the methodology are not limited to this case, and can be applied to other inactivation models and, even, other type of experiments (e.g. microbial growth).

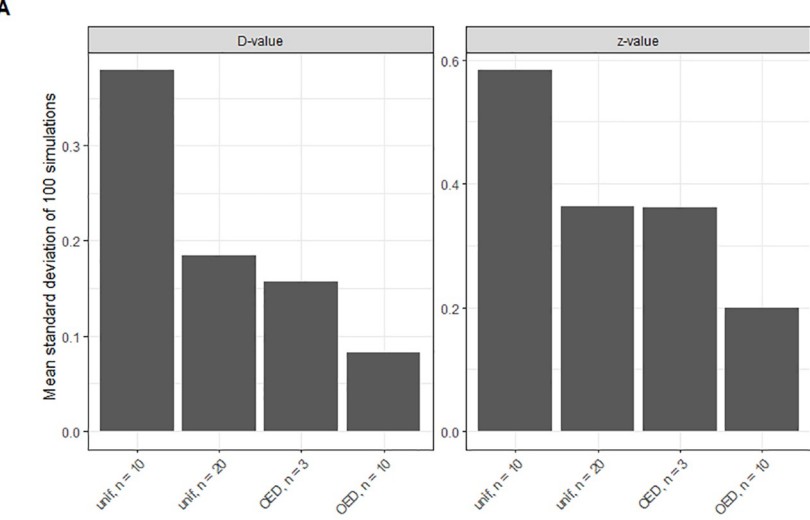

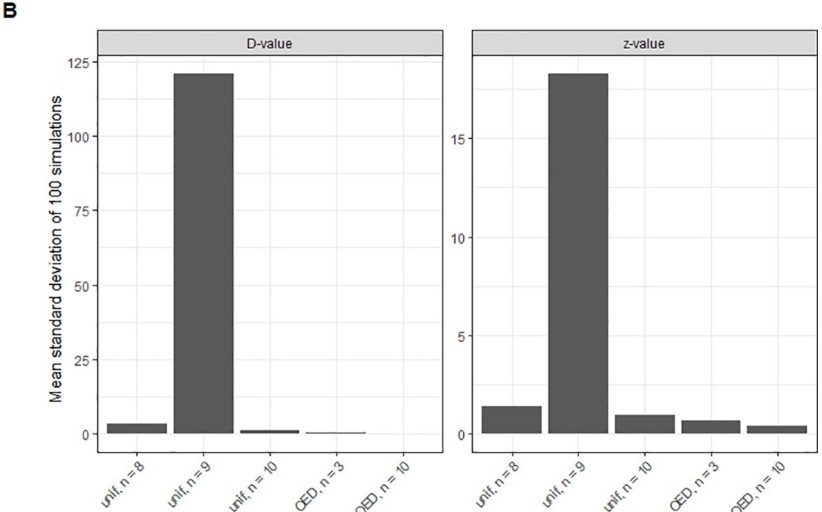

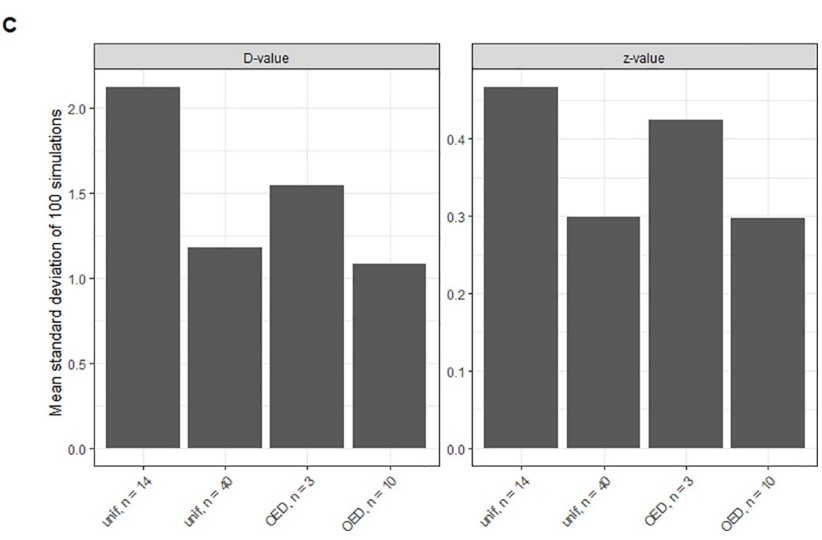

**Fig 5.** Comparison of the standard deviation estimated for the D and z-values for different experimental designs for the three thermal profiles analysed as case study (A, B and C).

## 4. Conclusions

The use of numerical simulations to aid on experimental designs has been illustrated in this work. Two complementary approaches have been proposed. The first, based on Monte Carlo simulations, can be used to assess whether an experimental design is likely to provide the desired level of precision. This approach is computationally intensive, so a second method, based on the properties of the FIM, has been suggested. The calculations required to apply it are affordable using current hardware and can easily be applied to compare between a large number of experimental designs. These methodologies enable to estimate whether an experimental design (with a selected number of sampling points) is likely to provide the desired precision in the parameter estimates (size of confidence intervals).

This methodology has been applied to compare between different experimental designs and inactivation profiles, identifying the most informative ones. The results highlight the advantages of OED with respect to uniform ones, attaining a higher precision in the parameter estimates in every situation tested. Moreover, the simulation results show that the addition of one sampling point in a uniform experimental design does not ensure an increase in accuracy. Hence, optimal designs are more robust and efficient than uniform designs. In spite of the overhead required for their calculation, they are recommended for researchers in food safety.

## Supporting information

**S1 Code. R code used for the comparison between experimental designs based on the FIM.**
(R)

**S2 Code. R code for the simulation of experiments in order to estimate the precision in parameter estimates.**
(R)

## Author Contributions

**Conceptualization:** Alberto Garre, Pablo S. Fernandez, Jose A. Egea.

**Formal analysis:** Alberto Garre, Pablo S. Fernandez, Jose A. Egea.

**Funding acquisition:** Alberto Garre, Pablo S. Fernandez.

**Investigation:** Alberto Garre, Jose Lucas Peñalver-Soto, Arturo Esnoz, Pablo S. Fernandez, Jose A. Egea.

**Methodology:** Alberto Garre, Jose Lucas Peñalver-Soto, Pablo S. Fernandez, Jose A. Egea.

**Project administration:** Pablo S. Fernandez, Jose A. Egea.

**Resources:** Asunción Iguaz, Pablo S. Fernandez.

**Software:** Alberto Garre, Jose Lucas Peñalver-Soto.

**Supervision:** Alberto Garre, Arturo Esnoz, Asunción Iguaz, Pablo S. Fernandez, Jose A. Egea.

**Validation:** Alberto Garre, Jose A. Egea.

**Visualization:** Jose A. Egea.

**Writing – original draft:** Alberto Garre.

**Writing – review & editing:** Alberto Garre, Jose Lucas Peñalver-Soto, Arturo Esnoz, Asunción Iguaz, Pablo S. Fernandez, Jose A. Egea.

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
