## [Decision Letter · Decision Letter 0]

14 Jun 2019

PONE-D-19-14978

On the use of in-silico simulations to support experimental design: a case study in microbial inactivation of foods

PLOS ONE

Dear Dr. Egea,

Thank you for submitting your manuscript to PLOS ONE. After careful consideration, we feel that it has merit but does not fully meet PLOS ONE’s publication criteria as it currently stands. Therefore, we invite you to submit a revised version of the manuscript that addresses the points raised during the review process.

Both reviewers agree that the manuscript needs major corrections.  Authors should pay special attention to critics regarding the methodology and data availability, as well as be clear on how data support the conclusions. Besides, although novelty and impact are not requisites in PLOS ONE, authors should be clear on how this work is original and different from the works in the literature. 

We would appreciate receiving your revised manuscript by Jul 29 2019 11:59PM. To enhance the reproducibility of your results, we recommend that if applicable you deposit your laboratory protocols in protocols.io, where a protocol can be assigned its own identifier (DOI) such that it can be cited independently in the future. For instructions see: http://journals.plos.org/plosone/s/submission-guidelines#loc-laboratory-protocols

We look forward to receiving your revised manuscript.

Kind regards,

Míriam R. García

Academic Editor

PLOS ONE

Journal Requirements:

Reviewers' comments:

Reviewer's Responses to Questions

**Comments to the Author**

1. Is the manuscript technically sound, and do the data support the conclusions?

Reviewer #1: Yes

Reviewer #2: Partly

2. Has the statistical analysis been performed appropriately and rigorously? 

Reviewer #1: Yes

Reviewer #2: No

3. Have the authors made all data underlying the findings in their manuscript fully available?

Reviewer #1: No

Reviewer #2: Yes

4. Is the manuscript presented in an intelligible fashion and written in standard English?

Reviewer #1: Yes

Reviewer #2: Yes

5. Review Comments to the Author

Reviewer #1: Summary:

The authors discuss the use of OED for identifying the Bigelow model for thermal microbial inactivation. Specifically, the manuscript focusses on the selection of the sampling times and the type of temperature profile.

Major comments:

- The statement on lines 88 and 89 is either not really correct or the authors did not clearly state what they meant. Since OED has been applied in predictive microbiology for over 20 years, it definitely falls already in the category of “currently available techniques”. As stated in the article, by simply calculating the inverse of the FIM for a proposed experimental design, it is already possible to have an idea of the precision that will be obtained on the parameter estimates (and this is just the underlying principle of OED). Therefore, currently available techniques can definitely provide such information.

- In general, the final section of the introduction (lines 88-100) should more clearly state the novelty of this research. Is it just providing information on optimal sampling schemes? Or is it the underlying method for determining such sampling schemes? Try to state more clearly what knowledge was already available and what is novel about this work with respect to the available research.

- The penalty function that is used (lines 153-162), appears to provide a weighting that is very arbitrary. Why don’t the authors apply constraints on the sampling times to be optimised? Using a set of linear inequality constraints could lead to a much less arbitrary solution to the same problem.

- For the results presented in Section 3.1, the sampling points of the OED designs should be illustrated and discussed. Use, e.g., a relative frequency bar chart to illustrate all sample points over the Monte Carlo simulation. These charts can be represented as three subfigures, corresponding with those of Figure 1. It should be possible to link these sampling points with the sensitivity equations in Figure 3.

- It is not really clear to me what the advantage is of the Monte Carlo method proposed in this publication. Why would you not just use the approximation of the variance-covariance matrix based on the inverse of the FIM of your design to estimate the uncertainty on the model parameters? What is the added value of this Monte Carlo simulation?

Minor comments:

- Line 21: Remove “)”.

- Figures 1 - 3: Use a line width of minimum 2 for the curves.

- Line 211: calculate -> calculating.

- Line 311: Takes what into account? Complete this sentence.

- Line 334-335: Use a capital letter for “Van Derlinden”.

- Line 345: smallest -> smaller.

- Simulation data should be made available.

Reviewer #2: The manuscript shows an interesting case study of microbial inactivation in foods with in-silico simulation analysis to support the suggestion of the two complementary methodologies to predict the parameter precision for a given experimental design. The manuscript is sound, but some major concerns are listed, and minor corrections are suggested below.

Major:

- Lines 115-121: The three profiles proposed have very different heating rates which impact on microbial inactivation. What assumptions were made to propose these temperature profiles? Can authors propose other temperature profiles, based on some thermal treatment of real food or other more realistic profiles for the case study (e.g. with residence time)? This is a concerning limitation issue, as suggested by own authors (lines 196-198) “Although previous studies have proposed algorithms to select optimal profiles, this example will be limited to the study of the three inactivation profiles shown in Figure 1”.

- Lines 144-145: Did authors test FIM criterions other than D-optimal? Why was D-optimal chosen? Advantages/disadvantages of that criterion against others in the context of the study (model and assumptions) should be presented.

- Lines 229-232: “One hundred Monte Carlo simulations have been performed, considering ten sampling points for each experimental design (OED and uniform) for each temperature profile. Simulations have been repeated with a higher number of simulations without observing differences in the results (not shown)”. Why the simulation tests started from ten sampling points? Kinetic inactivation experiments often have less than ten sampling points due to practical experimental issues. Furthermore, experiments should be simulated from less than ten sampling points in order to identify differences in the results (since no differences were reported with ten or more sampling points).

- Lines 249-251: “Therefore, the shape of the temperature profile should be taken into account when designing experiments for characterization of microbial inactivation under dynamic conditions”. Did authors try to design optimal temperature profiles before or together with optimal sampling points?

Minor:

- Lines 22-24. Present some reference about “scarce contributions”.

- Lines 78-81: FIM to quantify information in OED has been applied before Lehmann and Casella (1998).

- Line 120: 0.5 ºC/min and 10 ºC/min (and instead comma).

- Line 147: The “equation” (2) is incomplete (equal to?).

- Lines 183-226: Many information is about method and should be presented in appropriate section. Unnecessary repeated information can be removed. Results effectively start to be shown at line 231.

- Lines 242-243: “most of the inactivation occurs in a short time at the end of the experiment”. In Profile B most of the inactivation occurs at the middle of the experiment. In the end of the experiment, almost no inactivation occurs (due to the low temperature), as can be seen in Figure 1.

- Lines 298-318: Authors discussed about loss of information in uniform design measured by the inverse of the FIM determinant. In Figure 2C, there is an unexpected loss of information of OED experiment when adding from 16 to 17 points. How authors can explain this fact?

- Lines 374-375: “(3) that increasing the number of points in a uniform design does not ensure a higher precision”, as well as in some OED (e.g. profile C).

- Lines 471-472: Garre et al. 2017b reference is incomplete.

- Figure 3: The elements of the figure should have complete description in the caption. Information about A and B, red and green curves.

- Results of Monte Carlo simulations to D and z values could be used to presented and assess additional information.

6. PLOS authors have the option to publish the peer review history of their article (what does this mean?). If published, this will include your full peer review and any attached files.

Reviewer #1: Yes: Simen Akkermans

Reviewer #2: No

---

## [Author Response · Author response to Decision Letter 0]

27 Jun 2019

We would like to thank the reviewers and editor for their insightful comments on the paper, as they led us to improve its overall quality. Please find below our replies to the comments, including details about the corresponding changes incorporated in the new revised version of the manuscript.

Comments from the reviewers:

Reviewer #1:

Summary:

The authors discuss the use of OED for identifying the Bigelow model for thermal microbial inactivation. Specifically, the manuscript focusses on the selection of the sampling times and the type of temperature profile.

Major comments:

- The statement on lines 88 and 89 is either not really correct or the authors did not clearly state what they meant. Since OED has been applied in predictive microbiology for over 20 years, it definitely falls already in the category of “currently available techniques”. As stated in the article, by simply calculating the inverse of the FIM for a proposed experimental design, it is already possible to have an idea of the precision that will be obtained on the parameter estimates (and this is just the underlying principle of OED). Therefore, currently available techniques can definitely provide such information.

- In general, the final section of the introduction (lines 88-100) should more clearly state the novelty of this research. Is it just providing information on optimal sampling schemes? Or is it the underlying method for determining such sampling schemes? Try to state more clearly what knowledge was already available and what is novel about this work with respect to the available research.

We acknowledge both reviewer’s comments and agree in the fact that the novelty of this research was not correctly described in the original manuscript. We have rewritten the final part of the introduction addressing the two major comments.

“…The superiority of D-optimal designs with respect to uniform designs has been already demonstrated in several works (e.g. (Balsa-Canto et al., 2008; Garre et al., 2018b; Stamati et al., 2016).

“Despite the advances in OED, there is still high uncertainty in experimental design in the context of predictive microbiology. The most common goal of this type of experiments is the calculation of parameter estimates with a precision above a minimum (standard deviation of the model parameters). To date, there is not a clear methodology to estimate this. Therefore, some parameters of the design (e.g. the number of sampling points) is selected based on previous experience. Hence, there is a high risk that the number of sampling points is excessive, leading to unnecessary experimental work, or too low, which would require posterior repetitions of the experiment. In this work, we explore the application of numerical techniques to reduce this uncertainty. We propose two complementary methodologies, the first one based on the properties of the FIM and the second one based on Monte Carlo simulations. Although they have been applied in previous studies to compare between different designs, here we describe their mathematical basis and illustrate how they can be used to aid experimental design, reducing the risk of designs with excessive or too few sampling points. For this, we analyze a case study related to dynamic microbial inactivation. Nevertheless, the applicability of these methods is not restricted to this case and could, in principle, be applied to any problem in the context of predictive microbiology. “

- The penalty function that is used (lines 153-162), appears to provide a weighting that is very arbitrary. Why don’t the authors apply constraints on the sampling times to be optimised? Using a set of linear inequality constraints could lead to a much less arbitrary solution to the same problem.

We have used in this work the same penalization scheme that was used in Garre et al. (2018), where we explored different ways to describe the constraint. We agree that for most “classical” optimization algorithms linear inequality constraints are usually the best way to define this type of constraints. However, the optimization problem to be solved in the FIM-based OED requires a global optimization method which overcomes non-convexity issues. For that reason, we chose MEIGO which, as a metaheuristic, follows a very different strategy for the optimization. For this algorithm, linear inequality constraints were way less efficient (number of function evaluations) than the penalty function.

Nevertheless, we believe that a comparison of different ways to define the constraint is out of the scope of this research work. Note that our goal is the identification of experimental designs that are the most informative while, at the same time, being feasible from an experimental point of view. The way we have formulated the problem enables to calculate this in a reasonable time (some minutes) in a laptop (Windows 10, 1 core with 8 GB of RAM). More efficient formulations may exist for this particular problem, but exploring them would bring little benefit to the objective of this particular research work.

- For the results presented in Section 3.1, the sampling points of the OED designs should be illustrated and discussed. Use, e.g., a relative frequency bar chart to illustrate all sample points over the Monte Carlo simulation. These charts can be represented as three subfigures, corresponding with those of Figure 1. It should be possible to link these sampling points with the sensitivity equations in Figure 3.

We acknowledge the reviewer’s comment and agree that this is a relevant point that was not addressed in the original manuscript. As suggested, we have added a new subsection to the R&D section where the OEDs are described (section 3.1). It includes a new plot (Figure 2 in the new version). The new section now reads (L201-243):

3.1 Description of the Optimal Experiment Designs identified

Figure 2 gives a qualitative description of the designs identified as optimal for each one of the profiles analyzed (Figure 1), as well as on how they vary when the number of samples is increased. The x-axis represents the sample space (the duration of the experiment) and the height of the bars the number of samples that are located at a given location. The total number of sample points is represented by the colour of the bar (see legend in Figure 2). The bars corresponding to different number of points are stacked on top of each other, so the total height of the bar is a representation of the use of a time point across a different number of samples.

For every profile analyzed, some areas of the sampling space are the most informative and the algorithm tends to locate samples in that location. As expected, the most informative design pattern depends on the shape of the thermal profile. These differences between profiles can be justified based on the local sensitivity functions of each profile, shown in Figure 3. For profile A, samples are located in close to the end of the treatment, at approximately 56 and 60 minutes (Figure 2A). As illustrated in Figure 3A, t=56 corresponds to a minimum of the sensitivity function with respect to the z-value and t=60 to a supremum of both sensitivity functions. The algorithm is able to identify these areas and locates the sampling points in a configuration that satisfies the constraint related to the minimum distance between samples. For profile A, due to the large duration of the experiment (60 min) with respect to the minimum time between samples (3s), the restriction can be easily fulfilled. Hence, the frequency plot (Figure 2A) shows a large area without any samples between the two informative areas. 

Figure 2: Frequency bar plot illustrating the OED calculated for the three profiles analyzed (A, B and C). The colour of the bar indicates the total number of sampling points. 

Figure 3: Scaled local sensitivity functions of profile A (A), B (B) and C (C) with respect to the D-value (-) and the z-value (--).

Although profile C has a shape resembling the one of profile A, there are several differences that affect the optimal design pattern. Whereas the optimum design pattern for profile A distributes the samples in a balanced manner between t = 56 and t = 60, for profile C the end of the experiment is favoured. This is due to the fact that the sensitivity functions (especially the one of the z-value) grow quickly with temperature for temperatures about the reference one. The maximum temperature in profile C is 65�C, 5�C higher than the one reached in profile A. Consequently, the minimum of the sensitivity function with respect to the z-value is less relevant in profile C than in profile A. Furthermore, the duration of profile C is much lower than the one of profile A (2 min vs 60 min). Therefore, the constraint related to the minimum distance between sampling points is much harder to fulfil and the design is more spread-out. 

Profile B has a shape very different from those of profiles A and C. Hence, the optimal design pattern for this profile (Figure 2B) is very different to the one of the other two profiles. Several samples are located close to the middle of the treatment (t = 28 and t = 32). Both points correspond, respectively, to a minimum and a maximum of the sensitivity function corresponding to the z-value (Figure 3B). Besides this area, samples are located at the end of the experiment, where the sensitivity with respect of the D-value reaches its highest value. 

- It is not really clear to me what the advantage is of the Monte Carlo method proposed in this publication. Why would you not just use the approximation of the variance-covariance matrix based on the inverse of the FIM of your design to estimate the uncertainty on the model parameters? What is the added value of this Monte Carlo simulation?

The reviewer raises an interesting concern that was not properly addressed in the 1st version of the manuscript. The FIM has several shortcomings as an estimator of the variance-covariance matrix (C). First, according to the Cramer-Rao inequality, the FIM is a lower bound of the C. Moreover, the FIM does not take into account non-linearities in the model (10.3389/fbioe.2019.00122). Furthermore, the formulation of the FIM used in this work (as well as in most scientific works) is based on the hypothesis that the model is perfect and that errors are normal. 

The article has been expanded discussing this:

(L329): “On the other hand, the FIM has several shortcomings as an estimator of the variance-covariance matrix (C) of the model parameters. According to the Cramer-Rao inequality, the FIM is a bound of C under several hypotheses that disregard possible non-linearity in the model (Krausch et al., 2019). Moreover, the calculation of the FIM and the local sensitivities can be complicated when errors are non-normally distributed, for instance when residuals are heteroscedastic or when they do not follow a normal distribution (e.g. a Poisson distribution). Also, local sensitivity functions can be hard to calculate when the parameter of interest is the variance (e.g. in studies such as (Aryani et al., 2015)). Monte Carlo simulations are more flexible than the procedure based on the FIM are can be applied in such cases with little complexity.”

(L458): “Also, more complex hypotheses that are hard to implement in the FIM (e.g. heteroscedasticity of the residuals) can be implemented in the Monte Carlo simulations assess their impact. “

Minor comments:

- Line 21: Remove “)”.

Thank you for the comment. Corrected.

- Figures 1 - 3: Use a line width of minimum 2 for the curves.

The comment is acknowledged. The plots have been remade accordingly.

- Line 211: calculate -> calculating.

Thank you for the comment. Corrected.

- Line 311: Takes what into account? Complete this sentence.

We acknowledge the reviewer’s comment. The sentence has been rewritten as

“OED takes into account the amount of information when the number of sampling points is increased by one. On the other hand, uniform designs simply place the new point according to a uniform partition of the sampling space.”

- Line 334-335: Use a capital letter for “Van Derlinden”.

Thank you for the comment. Corrected.

- Line 345: smallest -> smaller.

Thank you for the comment. Corrected.

- Simulation data should be made available.

We agree with the reviewer’s comment. In the revised version, we have added the code used for the simulations as supplementary material.

Reviewer #2

The manuscript shows an interesting case study of microbial inactivation in foods with in-silico simulation analysis to support the suggestion of the two complementary methodologies to predict the parameter precision for a given experimental design. The manuscript is sound, but some major concerns are listed, and minor corrections are suggested below.

Major:

- Lines 115-121: The three profiles proposed have very different heating rates which impact on microbial inactivation. What assumptions were made to propose these temperature profiles? Can authors propose other temperature profiles, based on some thermal treatment of real food or other more realistic profiles for the case study (e.g. with residence time)? This is a concerning limitation issue, as suggested by own authors (lines 196-198) “Although previous studies have proposed algorithms to select optimal profiles, this example will be limited to the study of the three inactivation profiles shown in Figure 1”.

In lines 197-207, the parameters considered on the case study are described. They include the number of sampling points, their location and a selection between 3 different temperature profiles. We are analysing designs with between 3 and 20 sampling points. This means that we are comparing 108 different possible experimental designs (18 number of samples, uniform/OED, 3 profiles). Including the shape of the profile as an additional factor would increase very much the number of designs to be analysed without adding a significant information to the study.

Regarding the effect of the heating rates, this is a very interesting point. Their impact on the heating rates is still an active research field (e.g. 10.1016/j.ijfoodmicro.2017.11.023; 10.1016/j.foodcont.2012.05.042). Therefore, it is not possible to include this factor in this type of analysis. This hypothesis was already described in the original version of the manuscript (L122-129: “Although there is experimental evidence indicating otherwise…”) 

- Lines 144-145: Did authors test FIM criterions other than D-optimal? Why was D-optimal chosen? Advantages/disadvantages of that criterion against others in the context of the study (model and assumptions) should be presented.

We acknowledge the reviewer’s comment. We opted for using the D-criterion because it had been successfully used in similar problems. Being a relevant comment, we believe that comparison among optimality criterions is out of the scope of this research. The text has been modified discussing this point:

” Other criteria have been suggested to identify OEDs. A popular alternative is the E-criterion, which tries to minimize the maximum error (the model parameter with the highest error) (Balsa-Canto et al., 2008). Nevertheless, the D-criterion is preferred in some circumstances met in the case study presented here (Balsa-Canto et al., 2010) “

- Lines 229-232: “One hundred Monte Carlo simulations have been performed, considering ten sampling points for each experimental design (OED and uniform) for each temperature profile. Simulations have been repeated with a higher number of simulations without observing differences in the results (not shown)”. Why the simulation tests started from ten sampling points? Kinetic inactivation experiments often have less than ten sampling points due to practical experimental issues. Furthermore, experiments should be simulated from less than ten sampling points in order to identify differences in the results (since no differences were reported with ten or more sampling points).

We acknowledge the reviewer comment. The actual meaning of the paragraph (that we had made sure that the Monte Carlo algorithm had converged) was not clear in the original manuscript. We have rewritten that paragraph clarifying this topic.

“As described, the precision of the parameter estimates have been analysed using Monte Carlo simulations. In order to ensure the convergence of the algorithm, calculations have been repeated for different number of Monte Carlo simulations. Increasing their number beyond 100 simulations had no impact on the results (not shown), so the results obtained with 100 Monte Carlo simulations are reported.”

We have added a new subsection to the R&D section where the OEDs are described (section 3.1). This also helps to clarify the reviewer concern.

- Lines 249-251: “Therefore, the shape of the temperature profile should be taken into account when designing experiments for characterization of microbial inactivation under dynamic conditions”. Did authors try to design optimal temperature profiles before or together with optimal sampling points?

This point is, on its core, the same as MC#1. The shape of the temperature profile as a factor would very much increase the computational time without adding essential information to the results, which show that OED outperforms uniform designs in every case.

Minor:

- Lines 22-24. Present some reference about “scarce contributions”.

Although we appreciate the reviewer’s comment, the guidelines for PlosONE discourage the use of references in the abstract. In the introduction, this point is further discussed including references.

- Lines 78-81: FIM to quantify information in OED has been applied before Lehmann and Casella (1998).

We acknowledge the comment. The last part of the introduction has been rewritten clearly stating the novelty of the work.

- Line 120: 0.5 ºC/min and 10 ºC/min (and instead comma).

We appreciate the reviewer’s comment. Corrected.

- Line 147: The “equation” (2) is incomplete (equal to?).

We appreciate the reviewer’s comment. However, that is the standard way of writing an optimization problem (https://en.wikipedia.org/wiki/Optimization_problem).

- Lines 183-226: Many information is about method and should be presented in appropriate section. Unnecessary repeated information can be removed. Results effectively start to be shown at line 231.

We understand the reviewer comment. However, it must be understood that in this work analysis methods are not just a tool to analyse the data and reach conclusions. This research proposes the application of numerical methods to aid in the experimental design before any sampling point is taken. Hence, we believe that the discussion regarding how these can be actually applied and their limitations belongs to the R&D section, rather than to M&M. Future studies where this methodology is applied should describe it in M&M, but we believe that they belong to R&D in this work.

- Lines 242-243: “most of the inactivation occurs in a short time at the end of the experiment”. In Profile B most of the inactivation occurs at the middle of the experiment. In the end of the experiment, almost no inactivation occurs (due to the low temperature), as can be seen in Figure 1.

The reviewer’s comment is acknowledged. However, we are referring just to profile C in this sentence. It has been rewritten for clarification:

“…because most of the inactivation for profile C occurs in a short time (half a minute) at the end of the experiment”

- Lines 298-318: Authors discussed about loss of information in uniform design measured by the inverse of the FIM determinant. In Figure 2C, there is an unexpected loss of information of OED experiment when adding from 16 to 17 points. How authors can explain this fact?

The results shown in former Figure 2 (now Figure 4 after creating a new one and renumber the rest as suggested by Reviewer #1) require the application of several numerical methods (OED, sensitivity functions, determinant of the FIM, optimization) that introduce a numerical error (truncation and round-up) in the result. This is introducing some “noise” in the results shown in (former) Figure 2. Nonetheless, the OED remains much more informative than the uniform design. Moreover, according to the results of this investigation, increasing the number of sampling points for this profile beyond 12 samples is simply unreasonable. 

The text has been modified clarifying these points:

“…This impact is progressively diminished as the number of sampling points is increased. Indeed, it plateaus for profile C for more than 12 sampling points. This is caused by the constraint regarding the minimum distance between sampling points, that does not enable to increase the amount of information. Therefore, there is a limit in the amount of information that can be extracted for a thermal profile, when a restriction is included to limit the minimum distance between sampling points. For a number of samples beyond this limit, the inverse of the determinant of the FIM may slightly increase due to the numerical error involved in the calculations. Nonetheless, from the results in Figure 2C, it is unreasonable to increase the number of samples beyond 12 for profile C.”

- Lines 374-375: “(3) that increasing the number of points in a uniform design does not ensure a higher precision”, as well as in some OED (e.g. profile C).

This point is discussed in the previous minor correction.

- Lines 471-472: Garre et al. 2017b reference is incomplete.

We acknowledge the reviewer’s comment. It has been completed.

- Figure 3: The elements of the figure should have complete description in the caption. Information about A and B, red and green curves.

We understand that this information is already present in the legend: “Scaled local sensitivity functions of profile A (A) and B (B) with respect to the D-value (-) and the z-value (--).” “A” shows the sensitivity functions for profile A and “B” for profile B. The red curve is the sensitivity with respect to the D-value and green, dashed curve with respect to z.

- Results of Monte Carlo simulations to D and z values could be used to presented and assess additional information.

 We acknowledge the comment. The code for the simulations has been included as supplementary information, so that anyone can reproduce the calculations.

---

## [Decision Letter · Decision Letter 1]

9 Jul 2019

PONE-D-19-14978R1

On the use of in-silico simulations to support experimental design: a case study in microbial inactivation of foods

PLOS ONE

Dear Dr. Egea,

Thank you for submitting your manuscript to PLOS ONE. After careful consideration, we feel that the manuscript has improved considerably but still does not fully meet PLOS ONE’s publication criteria as it currently stands. Therefore, we invite you to submit a revised version of the manuscript that addresses the points raised during the review process.

Reviewer #2 has accepted the publication of the manuscript. However, previous reviewer #1 has declined the invitation for this second round, and we have had to invite a new reviewer with experience in the field. This new reviewer considers that there are still major concerns that should be clarified before publication. Also, please be more specific about the data availability. In the revised version of the manuscript we cannot find the supplementary material with the code or data used for the research.

We would appreciate receiving your revised manuscript by Aug 23 2019 11:59PM. To enhance the reproducibility of your results, we recommend that if applicable you deposit your laboratory protocols in protocols.io, where a protocol can be assigned its own identifier (DOI) such that it can be cited independently in the future. For instructions see: http://journals.plos.org/plosone/s/submission-guidelines#loc-laboratory-protocols

We look forward to receiving your revised manuscript.

Kind regards,

Míriam R. García

Academic Editor

PLOS ONE

Reviewers' comments:

Reviewer's Responses to Questions

**Comments to the Author**

1. If the authors have adequately addressed your comments raised in a previous round of review and you feel that this manuscript is now acceptable for publication, you may indicate that here to bypass the “Comments to the Author” section, enter your conflict of interest statement in the “Confidential to Editor” section, and submit your "Accept" recommendation.

Reviewer #2: All comments have been addressed

Reviewer #3: (No Response)

2. Is the manuscript technically sound, and do the data support the conclusions?

Reviewer #2: Yes

Reviewer #3: Partly

3. Has the statistical analysis been performed appropriately and rigorously? 

Reviewer #2: Yes

Reviewer #3: Yes

4. Have the authors made all data underlying the findings in their manuscript fully available?

Reviewer #2: Yes

Reviewer #3: No

5. Is the manuscript presented in an intelligible fashion and written in standard English?

Reviewer #2: Yes

Reviewer #3: No

6. Review Comments to the Author

Reviewer #2: (No Response)

Reviewer #3: The manuscript focusses on the use of Optimal Experimental Design (OED) to identify the most informative sampling schedule for the calibration of the Bigelow model for thermal microbial inactivation.

Despite an overall improvement can be detected in the second version of the manuscript, some of the concerns highlighted by previous reviewers have not been adequately addressed by the authors.

Major comments

- [Lines 105- 120]: The authors should clarify what to they mean by ‘there is still high uncertainty’ in the experimental designs used in predictive microbiology. Indeed the common tendency to set aspects of the experimental scheme adopted for model calibration based on past experience/technical limitations of the experimental system/acquisition platform applies to the proposed investigation (where only the effect of the number and location sampling times is considered, albeit not optimised). The authors should clearly state the reason for which they focus only on sampling times, as they could have investigated the effect of other aspects (e.g. the thermal perturbation profile). In addition, the use of scalar functions of the FIM to optimise experimental schemes is now routine. In this context, the actual contribution of this manuscript should be emphasised.

- The adopted formulation of the penalty function to set a constraint on the sampling frequency requires clarification. I understand that a similar formulation was adopted in a previous publication by the same authors, but i) hybrid solvers can cope with linear inequality constraints in FIM-based OED; ii) the optimisation cost cannot be too high in the considered case (1D model with 2 parameters).

- [Lines 167- 170]: E-criterion attempts to minimise the maximum uncertainty in parameter estimates (not the parameter with the highest error). In addition, the authors should clearly state which aspects of D-optimality meet ‘circumstances’ in their study case.

- Among the mentioned limitations of the FIM-based approach, local validity in the parameter space (i.e. computations are performed in the neighbourhood of the a priori unknown optimal/true parameters values) is the most relevant. I could not find any reference to how the initial uncertainty in parameter estimates was accounted for. This is a crucial point for applicability of the outlined methods. The application of the methods cannot precede the acquisition of some experimental data.

- An elusive comparison of the informative content of thermal profiles is presented. Due to the low dimensionality of the mathematical model, the simultaneous optimisation of i) number of sampling times, ii) their location and iii) the perturbation profile would have been feasible and of high interest.

- The conclusions drawn from the research provide limited insights and a more in depth analysis should be performed. It is intuitive that experience-based experimental schemes provide a lower bound of the performances of OED, otherwise we would not put effort in optimisation. The fact that one perturbation scheme results more informative than others is useful only if patterns identified support the extrapolation of ‘rules for informativeness’. Finally it seems obvious that data are not equally informative, so that increasing their cardinality might not convey additional information in uniform schemes.

Major comments

I report only some of the required corrections, please carefully revise the manuscript

- [Line 32]: methods to

- [Line 33]: is illustrated

- [Line 88]: the volume of the confidence hyperellipsoid

- [Line 91]: include additional ‘)’

- Figure 1 y axes ‘microbial’

- [Line 109]: are selected

- [Line 153]: criteria of the FIM. [… ], which consists

- [Line 178]: the selected parameter values

- [Line 181]: Hence, their effect

- [Line 213]: located close to..

- [Line 439]: an OED

- [Line 449]: different from

- [Line 463]: ‘more informative’

- [Lines 458-461]: Unclear, rephrase the sentence

- Figure 5: y axes ‘100 simulations’

7. PLOS authors have the option to publish the peer review history of their article (what does this mean?). If published, this will include your full peer review and any attached files.

Reviewer #2: No

Reviewer #3: Yes: Dr. Lucia Bandiera

---

## [Author Response · Author response to Decision Letter 1]

19 Jul 2019

Reviewer #3:

The manuscript focusses on the use of Optimal Experimental Design (OED) to identify the most informative sampling schedule for the calibration of the Bigelow model for thermal microbial inactivation.

Despite an overall improvement can be detected in the second version of the manuscript, some of the concerns highlighted by previous reviewers have not been adequately addressed by the authors.

Major comments

- [Lines 105- 120]: The authors should clarify what to they mean by ‘there is still high uncertainty’ in the experimental designs used in predictive microbiology. Indeed the common tendency to set aspects of the experimental scheme adopted for model calibration based on past experience/technical limitations of the experimental system/acquisition platform applies to the proposed investigation (where only the effect of the number and location sampling times is considered, albeit not optimised). The authors should clearly state the reason for which they focus only on sampling times, as they could have investigated the effect of other aspects (e.g. the thermal perturbation profile). In addition, the use of scalar functions of the FIM to optimise experimental schemes is now routine. In this context, the actual contribution of this manuscript should be emphasised.

We acknowledge the reviewer’s comment. By “high uncertainty” we refer to a very common question in any microbiology laboratory: “how many sampling points do I need?” This can refer, for instance, to the number of sampling points taken in an inactivation experiment for characterizing the microbial response. This is usually decided based on experience. As a result, there is a high risk of designing experiments with too many points (resulting in additional experimental work) or with too few (requiring a repetition of the experiment). In this article, we propose the application of numerical techniques as a science-based approach to aid in experimental design.

In this research we optimize the position of sampling times. We apply the methodology for OED we previously developed (Garre et al., 2018b) to identify optimal sampling schemes (Section 2.2; Figure 2). Including the shape of the thermal profile as a parameter of the experimental design is an interesting exercise from the point of view of the experimental design that was studied in a previous research work (van Derlinden et al., 2010). However, we have some concerns regarding that approach from a biological point of view. It has been demonstrated that the shape of the thermal profile (i.e. the heating rate) can affect the microbial response. If the heating is slow, a physiological response can be triggered in the cells that increases their stress resistance (Corradini and Peleg, 2009; Dolan et al., 2013; Garre et al., 2018c, 2018a; Hassani et al., 2007, 2006; Stasiewicz et al., 2008; Valdramidis et al., 2007). Fast heating profiles can also influence the bacterial response to stress (de Jong et al., 2012; de Jonge, 2019; Huertas et al., 2016). The model-based approach for OED applied here is based on the hypothesis that the design does not affect the validity of the model (i.e. of the model parameters). We understand that including the shape of the temperature profile as a variable in the optimization problem collides with this hypothesis, due to the experimental evidence which indicates that it affects the microbial response. Nevertheless, the effect of heating rates on microbial inactivation is still an active research topic. A detailed discussion of this matter would take the focus out of the main research question addressed in the article, so it has not been included in the manuscript. 

We disagree with the claim that OED is common practice in predictive microbiology. Although it is more commonly used in other fields (e.g. systems biology), uniform sampling schemes are still the default approach for describing microbial growth and/or inactivation under non-isothermal conditions (Conesa et al., 2009; Corradini and Peleg, 2004; Franco-Vega et al., 2015; Haberbeck et al., 2013; Huang, 2013; Huertas et al., 2015; Mattick et al., 2001; Ros-Chumillas et al., 2015; Valdramidis et al., 2008). We firmly believe that OEDs have several advantages with respect to uniform designs, but they must be illustrated with case studies before they are broadly accepted by the community (especially scientists without a strong background in statistics). This article serves that purpose, showing the drawbacks of uniform designs: lower precision for the same experimental effort, and the fact that they not ensure that increasing the number of sampling points reduces parameter uncertainty.

The last paragraph of the introduction has been largely rewritten including these remarks:

“

The model parameters of models used in predictive microbiology usually have a biological meaning. For instance, the D-value describes the treatment time required to reduce the microbial count a 90% (Bigelow, 1921). Model parameters estimated under certain conditions are commonly used to, for example, infer the effectiveness of a treatment (Maté et al., 2016; Ros-Chumillas et al., 2017). Therefore, in many situations, the objective of experiments designed in the context of predictive microbiology is not prediction but the estimation of model parameter with enough precisión (standard deviation) that enables accurate inference. Despite the advances in OED, there are still some open questions when it comes to designing such experiments. For instance, the number of sampling points is commonly decided based on previous experience. As a result, there is a high risk that the number of sampling points is excessive, leading to unnecessary experimental work, or too low, which would require posterior repetitions of the experiment. In this work, we explore the application of numerical techniques to reduce this uncertainty. We propose two complementary methodologies, the first one based on the properties of the FIM and the second one based on Monte Carlo simulations. Although both methods are usually applied to compare between different designs, here we illustrate how they can be used to aid in the decision process during the first stages of the experimental design. We describe their mathematical basis and illustrate how they can provide valuable information that may reduce the uncertainty of the experimental design (e.g. in the selection of the number of sampling points). For this, we analyze a case study related to dynamic microbial inactivation. Nevertheless, the applicability of these methods is not restricted to this case and could, in principle, be applied to any problem in the context of predictive microbiology. 

“

- The adopted formulation of the penalty function to set a constraint on the sampling frequency requires clarification. I understand that a similar formulation was adopted in a previous publication by the same authors, but i) hybrid solvers can cope with linear inequality constraints in FIM-based OED; ii) the optimisation cost cannot be too high in the considered case (1D model with 2 parameters).

We appreciate the reviewer’s suggestion. In a previous investigation (Garre et al., 2018b), we tried different formulations of the optimization problem and we found that the addition of the penalty function provided feasible solutions which could not be improved by applying a final local search with a local method (i.e., interior point method) which handles non-linear constraints. In this research, we have used a global optimization method, as recommended by different authors to address this type of problem (e.g., Balsa-Canto et al. 2008). We also tried another global optimization method handling inequality constraints, the Improved Stochastic Ranking Evolution Strategy (ISRES), implemented in the nloptr R package (https://cran.r-project.org/web/packages/nloptr/index.html) and after much longer runs than those performed with MEIGO and the mentioned penalty function, the obtained solutions were worse than those presented in our work. We also used the Rsolnp method (https://cran.r-project.org/web/packages/Rsolnp/index.html), which is a general non-linear optimization method using augmented Lagrange multipliers and, again, the results did not outperformed those obtained with the proposed formulation/method. The best solutions were provided by the enhanced scatter search method implemented in MEIGO using a final local refinement with a direct search method based on hill climbing (DHC). The package bioOED referenced in our paper allows users to use MEIGO with its different options and local solvers to solve OED problems in predictive microbiology.

We are aware that we do not have a mathematical prove that the obtained solutions are optimal, but we obtain better solutions than other alternatives, we use the recommended type of solve for this type of problem, and we believe that the possible difference between the provided solutions and the mathematically optimal solutions is probably due to a matter of convergence tolerance in the decision variables and/or objective function. 

We agree with the reviewer that there is the possibility that a smarter mathematical formulation of the optimization problem exists. However, although the model is relatively simple, the optimization is done according to the position of each sampling point. Therefore, an experimental design with 18 sampling points has 18 decision variables. We believe that a deeper study of the optimization problem, although interesting from the point of view of optimization theory, would only have a small contribution to the research question (how the properties of the FIM and MC simulations can aid in experimental design) and would increase the complexity of the analysis.

- [Lines 167- 170]: E-criterion attempts to minimise the maximum uncertainty in parameter estimates (not the parameter with the highest error). In addition, the authors should clearly state which aspects of D-optimality meet ‘circumstances’ in their study case.

We acknowledge the reviewer’s comment. We have referred to the E-criterion in the text to make the reader aware of the fact that there are other alternatives. However, a variety of criteria have been suggested for the quantification of the information observable using an experimental design (A-criterion, C-criterion, D-criterion, E-criterion, T-criterion, G-criterion, I-criterion…) and many of them even have modified version (e.g. the A-criterion) with better properties from the point of view of the optimization problem. As well as with the previous comment, we believe that a comparison between the designs identified using each one of the criteria listed, although very interesting from the point of view of experimental design theory, would only bring a small contribution to the research question analysed. Hence, it is left for future work. In any case, the R package bioOED with which the calculations can be reproduced, allows the user to switch between D and modified E-criterion to explore other possible designs.

According to the reviewer’s remarks and the paragraph has been rephrased including additional information:

“A popular alternative is the E-criterion, which tries to minimize the maximum uncertainty in parameter estimates (in the case studied here, the uncertainty of the parameter with the highest error) (Balsa-Canto et al., 2008). Because this article tries to illustrate how computational methods can be used to aid experimental design and because this criterion has already been applied in a similar problem (Garre et al., 2018b), this study is limited to the results of the D-criterion. A comparison between the precision of different designs is left for future work.”

- Among the mentioned limitations of the FIM-based approach, local validity in the parameter space (i.e. computations are performed in the neighbourhood of the a priori unknown optimal/true parameters values) is the most relevant. I could not find any reference to how the initial uncertainty in parameter estimates was accounted for. This is a crucial point for applicability of the outlined methods. The application of the methods cannot precede the acquisition of some experimental data.

We acknowledge the reviewer’s comment. The need for nominal parameters values is a known limitation of the approach to OED based on the FIM. We have used model parameters taken from the literature for experimental conditions similar to the ones considered in our study. This approach has also been applied in other articles that also studied OED in the context of predictive microbiology (Akkermans et al., 2018b, 2018a; Bernaerts et al., 2000; Garre et al., 2018b; Poschet et al., 2005; Stamati et al., 2016; van Derlinden et al., 2010; Versyck et al., 1999, 1997). Due to the availability of extensive reviews and databases of microbial inactivation (Baranyi and Tamplin, 2004; Doyle et al., 2001; Doyle and Mazzotta, 2000; van Asselt and Zwietering, 2006), approximate values of the model parameters can be found for most conditions.

In a practical setting, the experimental design could be updated after the first repetition. This approach, quite common in, for instance, systems biology, is usually not required in predictive microbiology, due to the lower dimensionality of the models.

In any case we find the reviewer remark very interesting and we will consider performing a new study based on global sensitivity analysis to guide predictive microbiologist in the uncommon cases where nominal values for the parameters cannot be found anywhere. 

- An elusive comparison of the informative content of thermal profiles is presented. Due to the low dimensionality of the mathematical model, the simultaneous optimisation of i) number of sampling times, ii) their location and iii) the perturbation profile would have been feasible and of high interest.

See response to the first major comment. We agree with the reviewer that finding optimal temperature profiles for designing optimal experiments or in other tasks like process design and optimization is a very relevant engineering question. However, as mentioned above, the temperature profile in this context is crucial for biological aspects like the microorganism acclimation. In such cases, the model that best describes the microbial counts within a thermal inactivation process would be different, with a different number of parameters. We have preferred to let the temperature profile a as discrete given factor (i.e., by considering 3 different profiles which could have been extended to other commonly used in research or industry) to illustrate the main objective of the paper: OED + MC simulations is an effective methodology to select the sampling points in thermal microbial inactivation and get the maximum information.

- The conclusions drawn from the research provide limited insights and a more in depth analysis should be performed. It is intuitive that experience-based experimental schemes provide a lower bound of the performances of OED, otherwise we would not put effort in optimisation. The fact that one perturbation scheme results more informative than others is useful only if patterns identified support the extrapolation of ‘rules for informativeness’. Finally it seems obvious that data are not equally informative, so that increasing their cardinality might not convey additional information in uniform schemes.

We acknowledge the reviewer’s comment. She raises several objection that we will respond one by one.

We agree that experience-based are a lower bound for OED. That’s why we always try to include uniform designs as a baseline for comparison with the OED (the definition of an “experience-based” one can be researcher-dependent). We have found some patterns of perturbation, shown in Figure 2, that are dependent on the temperature profile and are stable with the number of sampling points. However, we have not included the shape of the thermal profile in those patterns due to the biological arguments we have given in our response to the previous comments.

Regarding the uniform design, an expert on experiment design may find this point obvious. But we believe that researchers with a background that is more focused on microbiology expect that increasing the number of sampling points would reduce the uncertainty in parameter estimates. Indeed, it is very rare for articles in predictive microbiology to justify the number of sampling points. In this research work we clearly illustrate that increasing the number of sampling points does not ensure that the uncertainty in parameter estimates is reduced. We could not find such result published before in the context of predictive microbiology and we believe that some researchers may find this point, at least, surprising. We believe this is an interesting research finding, which can encourage predictive microbiologists with a background in microbiology to apply OEDs.

Our goal with this work was to illustrate how computational tools (the properties of the FIM and Monte Carlo simulations) can aid in experimental design. It can be used to compare between different designs (optimal/uniform, as well as between different profiles) and to evaluate how increasing the number of sampling points affects uncertainty. The results show that for each thermal profile, there is an upper bound for the amount of information that can be extracted, so uncertainty cannot be reduced below a minimum. This is already an interesting result for predictive microbiology, where it is usually assumed that uncertainty can be reduced to zero if enough information (experimental data, in this case) is gathered. Furthermore, we demonstrate that increasing the number of sampling points may actually increase uncertainty when a uniform sampling scheme is used. Finally, we illustrate how Monte Carlo simulations can be used to decide the number of sampling points in an experiment (maximum standard deviation of the parameters). To date, that was done based on previous experience, so there was a high risk of taking too many or too few points.

We believe that these conclusions are relevant for predictive microbiology and that have enough merit to be published.

Minor comments

We thank the reviewer for reporting the list of found typos and errors. Every minor correction has been applied as suggested. Furthermore, the whole manuscript has been proof-read and several typos have been corrected.

- [Lines 458-461]: Unclear, rephrase the sentence

The sentence has been rephrased as: 

“Also, some typical model assumptions (e.g. homogeneity of the residuals) can be checked by means of the Monte Carlo simulations”

References

Akkermans, S., Logist, F., Van Impe, J.F., 2018a. Parameter estimations in predictive microbiology: Statistically sound modelling of the microbial growth rate. Food Res. Int. 106, 1105–1113. https://doi.org/10.1016/j.foodres.2017.11.083

Akkermans, S., Nimmegeers, P., Van Impe, J.F., 2018b. Comparing design of experiments and optimal experimental design techniques for modelling the microbial growth rate under static environmental conditions. Food Microbiol. https://doi.org/10.1016/j.fm.2018.05.010

Balsa-Canto, E., Alonso, A.A., Banga, J.R., 2008. Computational procedures for optimal experimental design in biological systems. IET Syst. Biol. 2, 163–172. https://doi.org/10.1049/iet-syb:20070069

Baranyi, J., Tamplin, M.L., 2004. ComBase: A Common Database on Microbial Responses to Food Environments. J. Food Prot. 67, 1967–1971. https://doi.org/10.4315/0362-028X-67.9.1967

Bernaerts, K., Versyck, K.J., Van Impe, J.F., 2000. On the design of optimal dynamic experiments for parameter estimation of a Ratkowsky-type growth kinetics at suboptimal temperatures. Int. J. Food Microbiol. 54, 27–38.

Conesa, R., Andreu, S., Fernández, P.S., Esnoz, A., Palop, A., 2009. Nonisothermal heat resistance determinations with the thermoresistometer Mastia. J. Appl. Microbiol. 107, 506–513. https://doi.org/10.1111/j.1365-2672.2009.04236.x

Corradini, M.G., Peleg, M., 2009. Dynamic Model of Heat Inactivation Kinetics for Bacterial Adaptation. Appl. Environ. Microbiol. 75, 2590–2597. https://doi.org/10.1128/AEM.02167-08

Corradini, M.G., Peleg, M., 2004. Demonstration of the applicability of the Weibull–log-logistic survival model to the isothermal and nonisothermal inactivation of Escherichia coli K-12 MG1655. J. Food Prot. 67, 2617–2621.

de Jong, A.E.I., van Asselt, E.D., Zwietering, M.H., Nauta, M.J., de Jonge, R., 2012. Extreme Heat Resistance of Food Borne Pathogens Campylobacter jejuni, Escherichia coli, and Salmonella typhimurium on Chicken Breast Fillet during Cooking [WWW Document]. Int. J. Microbiol. https://doi.org/10.1155/2012/196841

de Jonge, R., 2019. Predictable and unpredictable survival of foodborne pathogens during non-isothermal heating. Int. J. Food Microbiol. 291, 151–160. https://doi.org/10.1016/j.ijfoodmicro.2018.11.018

Dolan, K.D., Valdramidis, V.P., Mishra, D.K., 2013. Parameter estimation for dynamic microbial inactivation: which model, which precision? Food Control, Predictive Modelling of Food Quality and Safety 29, 401–408. https://doi.org/10.1016/j.foodcont.2012.05.042

Doyle, M.E., Mazzotta, A.S., 2000. Review of Studies on the Thermal Resistance of Salmonellae. J. Food Prot. 63, 779–795. https://doi.org/10.4315/0362-028X-63.6.779

Doyle, M.E., Mazzotta, A.S., Wang, T., Wiseman, D.W., Scott, V.N., 2001. Heat Resistance of Listeria monocytogenes. J. Food Prot. 64, 410–429. https://doi.org/10.4315/0362-028X-64.3.410

Franco-Vega, A., Ramírez-Corona, N., López-Malo, A., Palou, E., 2015. Estimation of Listeria monocytogenes survival during thermoultrasonic treatments in non-isothermal conditions: Effect of ultrasound on temperature and survival profiles. Food Microbiol. 52, 124–130. https://doi.org/10.1016/j.fm.2015.07.006

Garre, A., Egea, J.A., Iguaz, A., Palop, A., Fernandez, P.S., 2018a. Relevance of the Induced Stress Resistance When Identifying the Critical Microorganism for Microbial Risk Assessment. Front. Microbiol. 9. https://doi.org/10.3389/fmicb.2018.01663

Garre, A., González-Tejedor, G., Peñalver-Soto, J.L., Fernández, P.S., Egea, J.A., 2018b. Optimal characterization of thermal microbial inactivation simulating non-isothermal processes. Food Res. Int. 107, 267–274. https://doi.org/10.1016/j.foodres.2018.02.040

Garre, A., Huertas, J.P., González-Tejedor, G.A., Fernández, P.S., Egea, J.A., Palop, A., Esnoz, A., 2018c. Mathematical quantification of the induced stress resistance of microbial populations during non-isothermal stresses. Int. J. Food Microbiol. 266, 133–141. https://doi.org/10.1016/j.ijfoodmicro.2017.11.023

Haberbeck, L.U., Dannenhauer, C., Salomão, B.D.C.M., De Aragão, G.M.F., 2013. Estimation of the Thermochemical Nonisothermal Inactivation Behavior of Bacillus Coagulans Spores in Nutrient Broth with Oregano Essential Oil. J. Food Process. Preserv. 37, 962–969. https://doi.org/10.1111/j.1745-4549.2012.00745.x

Hassani, M., Cebrián, G., Mañas, P., Condón, S., Pagán, R., 2006. Induced thermotolerance under nonisothermal treatments of a heat sensitive and a resistant strain of Staphylococcus aureus in media of different pH. Lett. Appl. Microbiol. 43, 619–624. https://doi.org/10.1111/j.1472-765X.2006.02014.x

Hassani, M., Manas, P., Pagán, R., Condón, S., 2007. Effect of a previous heat shock on the thermal resistance of Listeria monocytogenes and Pseudomonas aeruginosa at different pHs. Int. J. Food Microbiol. 116, 228–238.

Huang, L., 2013. Determination of thermal inactivation kinetics of Listeria monocytogenes in chicken meats by isothermal and dynamic methods. Food Control 33, 484–488. https://doi.org/10.1016/j.foodcont.2013.03.049

Huertas, J.-P., Aznar, A., Esnoz, A., Fernández, P.S., Iguaz, A., Periago, P.M., Palop, A., 2016. High Heating Rates Affect Greatly the Inactivation Rate of Escherichia coli. Front. Microbiol. 7. https://doi.org/10.3389/fmicb.2016.01256

Huertas, J.-P., Ros-Chumillas, M., Esteban, M.-D., Esnoz, A., Palop, A., 2015. Determination of Thermal Inactivation Kinetics by the Multipoint Method in a Pilot Plant Tubular Heat Exchanger. Food Bioprocess Technol. 8, 1543–1551. https://doi.org/10.1007/s11947-015-1525-9

Mattick, K.L., Legan, J.D., Humphrey, T.J., Peleg, M., 2001. Calculating Salmonella inactivation in nonisothermal heat treatments from isothermal nonlinear survival curves. J. Food Prot. 64, 606–613.

Poschet, F., Geeraerd, A.H., Van Loey, A.M., Hendrickx, M.E., Van Impe, J.F., 2005. Assessing the optimal experiment setup for first order kinetic studies by Monte Carlo analysis. Food Control 16, 873–882. https://doi.org/10.1016/j.foodcont.2004.07.009

Ros-Chumillas, M., Esteban, M.-D., Huertas, J.-P., Palop, A., 2015. Effect of Nisin and Thermal Treatments on the Heat Resistance of Clostridium sporogenes Spores. J. Food Prot. 78, 2019–2023.

Stamati, I., Akkermans, S., Logist, F., Noriega, E., Van Impe, J., 2016. Optimal experimental design for discriminating between microbial growth models as function of suboptimal temperature: From in silico to in vivo. Food Res. Int. 89, 689–700. https://doi.org/10.1016/j.foodres.2016.08.001

Stasiewicz, M.J., Marks, B.P., Orta-Ramirez, A., Smith, D.M., 2008. Modeling the Effect of Prior Sublethal Thermal History on the Thermal Inactivation Rate of Salmonella in Ground Turkey. J. Food Prot. 71, 279–285. https://doi.org/10.4315/0362-028X-71.2.279

Valdramidis, V.P., Geeraerd, A.H., Bernaerts, K., Van Impe, J.F.M., 2008. Identification of non-linear microbial inactivation kinetics under dynamic conditions. Int. J. Food Microbiol. 128, 146–152.

Valdramidis, V.P., Geeraerd, A.H., Van Impe, J.F., 2007. Stress-adaptive responses by heat under the microscope of predictive microbiology: Modelling the microbial heat resistance. J. Appl. Microbiol. 103, 1922–1930. https://doi.org/10.1111/j.1365-2672.2007.03426.x

van Asselt, E.D., Zwietering, M.H., 2006. A systematic approach to determine global thermal inactivation parameters for various food pathogens. Int. J. Food Microbiol. 107, 73–82. https://doi.org/10.1016/j.ijfoodmicro.2005.08.014

van Derlinden, E., Balsa-Canto, Van Impe, 2010. (Optimal) experiment design for microbial inactivation, in: Progress on Quantitative Approaches of Thermal Food Processing - Valdramidis, V.P. & Van Impe, J.F.M. (Eds), Advances in Food Safety and Food Microbiology. Nova Publishers, pp. 67–98.

Versyck, K.J., Bernaerts, K., Geeraerd, A.H., Van Impe, J.F., 1999. Introducing optimal experimental design in predictive modeling: A motivating example. Int. J. Food Microbiol. 51, 39–51. https://doi.org/10.1016/S0168-1605(99)00093-8

Versyck, K.J., Claes, J.E., Impe, J.F.V., 1997. Practical Identification of Unstructured Growth Kinetics by Application of Optimal Experimental Design. Biotechnol. Prog. 13, 524–531. https://doi.org/10.1021/bp970080j

---

## [Decision Letter · Decision Letter 2]

23 Jul 2019

On the use of in-silico simulations to support experimental design: a case study in microbial inactivation of foods

PONE-D-19-14978R2

Dear Dr. Egea,

We are pleased to inform you that your manuscript has been judged scientifically suitable for publication and will be formally accepted for publication once it complies with all outstanding technical requirements.

With kind regards,

Míriam R. García

Academic Editor

PLOS ONE

Additional Editor Comments (optional):

Reviewers' comments:

Reviewer's Responses to Questions

**Comments to the Author**

1. If the authors have adequately addressed your comments raised in a previous round of review and you feel that this manuscript is now acceptable for publication, you may indicate that here to bypass the “Comments to the Author” section, enter your conflict of interest statement in the “Confidential to Editor” section, and submit your "Accept" recommendation.

Reviewer #3: All comments have been addressed

2. Is the manuscript technically sound, and do the data support the conclusions?

Reviewer #3: Yes

3. Has the statistical analysis been performed appropriately and rigorously? 

Reviewer #3: Yes

4. Have the authors made all data underlying the findings in their manuscript fully available?

Reviewer #3: Yes

5. Is the manuscript presented in an intelligible fashion and written in standard English?

Reviewer #3: Yes

6. Review Comments to the Author

Reviewer #3: (No Response)

7. PLOS authors have the option to publish the peer review history of their article (what does this mean?). If published, this will include your full peer review and any attached files.

Reviewer #3: Yes: Lucia Bandiera

---

## [Editor Report · Acceptance letter]

15 Aug 2019

PONE-D-19-14978R2 

On the use of in-silico simulations to support experimental design: a case study in microbial inactivation of foods 

Dear Dr. Egea:

I am pleased to inform you that your manuscript has been deemed suitable for publication in PLOS ONE. Congratulations! Your manuscript is now with our production department. 

With kind regards,

on behalf of

Dr. Míriam R. García 

Academic Editor

PLOS ONE